# Is Automated Topic Model Evaluation Broken?: The Incoherence of Coherence

**Alexander Hoyle**[*]    **Pranav Goel**[*]    **Denis Peskov**[*]    **Andrew Hian-Cheong**[*]
Computer Science

**Jordan Boyd-Graber**        **Philip Resnik**
CS, iSchool, UMIACS, LSC    UMIACS, Lingusitics

University of Maryland
{hoyle,pgoel1,dpeskov,andrewhc,jbg,resnik}@cs.umd.edu

## Abstract

Topic model evaluation, like evaluation of other unsupervised methods, can be contentious. However, the field has coalesced around automated estimates of topic coherence, which rely on the frequency of word co-occurrences in a reference corpus. Contemporary neural topic models surpass classical ones according to these metrics. At the same time, topic model evaluation suffers from a *validation gap*: automated coherence, developed for classical models, has not been validated using human experimentation for neural models. In addition, a meta-analysis of topic modeling literature reveals a substantial *standardization gap* in automated topic modeling benchmarks. To address the validation gap, we compare automated coherence with the two most widely accepted human judgment tasks: topic rating and word intrusion. To address the standardization gap, we systematically evaluate a dominant classical model and two state-of-the-art neural models on two commonly used datasets. Automated evaluations declare a winning model when corresponding human evaluations do not, calling into question the validity of fully automatic evaluations independent of human judgments.

## 1 Revisiting Topic Model Evaluation

Topic models are a machine learning technique widely used outside computer science, including political science (Grimmer and Stewart, 2013; Isoaho et al., 2021), social and cultural studies (Mohr and Bogdanov, 2013), digital humanities (Meeks and Weingart, 2012), and bioinformatics (Liu et al., 2016). Typically, topic model users are domain experts trying to identify global categories or themes present in a document collection (Boyd-Graber et al., 2017). This practice constitutes a computer-assisted form of content analysis (Krippendorff, 2004; Chuang et al., 2014), also related to distant reading in literary studies (Underwood, 2017). In general, topic models help humans understand large corpora.[2]

Evaluation of topic models has vacillated between automated and human-centered. While real-world users of topic models evaluate outputs based on their specific needs, topic model developers have gravitated toward generalized, automated proxies of human judgment to help inform rapid iteration of models (Doogan and Buntine, 2021). Initially, models were evaluated with held-out perplexity, but it disagrees with human interpretability (Chang et al., 2009). Consequently, the field adopted automated coherence metrics like normalized pointwise mutual information (NPMI), a measure of word relatedness that *does* correlate with topic interpretability (Section 2.2; Newman et al., 2010; Aletras and Stevenson, 2013; Lau et al., 2014). The balance shifted towards automated coherence.

Human evaluations have been abandoned by topic model developers in the years since automated coherence metrics were adopted. In a thorough meta-analysis of contemporary topic model methods

---

[*]Equal contribution

[2]Topic models are also used for other purposes, such as information retrieval or downstream document classification. However, the discovery and application of categories for human interpretation is their dominant use, and other computational applications have been largely eclipsed by modern neural approaches.

35th Conference on Neural Information Processing Systems (NeurIPS 2021).

|  | **Classical** | | | **Neural** | | |
|---|---|---|---|---|---|---|
|  | station | album | tropical | tropical | spore | manhattan_project |
|  | line | band | storm | landfall | basidia | los_alamos_laboratory |
|  | bridge | music | hurricane | cyclone | spores | robert_oppenheimer |
|  | railway | song | cyclone | utc | mycologist | enrico_fermi |
|  | trains | released | depression | weakening | hyphae | physicist |
| **NPMI** | 0.274 | 0.285 | 0.394 | 0.446 | 0.456 | 0.470 |

Table 1: The first three columns are the highest-NPMI topics for a classical topic model (LDA estimated via Gibbs sampling using Mallet, McCallum, 2002; Griffiths and Steyvers, 2004). The next three are counterparts from a neural model (our D-VAE reimplementation, Burkhardt and Kramer, 2019). Models are trained on Wikitext (Merity et al., 2017) with fifty topics, and NPMI is estimated over the top five words in each topic using a 4.6M-document reference Wikipedia corpus. The mean top-five NPMI over all topics is 0.156 for the classical and 0.256 for the neural model.

papers, *none* conduct systematic human evaluations (Section 3). Instead, they rely solely on automated metrics for model comparison.[3] However, current neural topic models are a far cry from the classical models that substantiated the original correlations—manifestly, topics produced by neural models are often qualitatively distinct from those of classical models (e.g., Table 1).[4] This *validation gap* raises the question of whether automated metrics are still consistent with human judgments of topic quality.

Moreover, we should always be cautious when extrapolating outside the range of data that was used to establish a relationship between variables. As an example, a neural model in Hoyle et al. (2020) produces much larger NPMI values than those used to determine human correlations in the original Lau et al. (2014) study; the implicit assumption is that greater NPMI corresponds to more human-interpretable topics. Finally, a myopic focus on a presumed proxy for human preferences can produce low-quality results (Stiennon et al., 2020). Does Goodharts' law—"when a measure becomes a target, it ceases to be a good measure" (Strathern, 1997)—apply to automated metrics of topic models?

Another challenge for automated evaluation, whether of classical or neural topic models, is widespread inconsistency (Section 3). Researchers frequently fail to specify the information needed to calculate automated metrics or diverge from the practices that underpin human correlations. Furthermore, evaluation datasets, preprocessing, and hyperparameter optimization vary dramatically, even within a given paper. This *standardization gap* likely limits the generalizability and reliability of topic model developers' findings.

We address the standardization and validation gaps in topic model evaluation:

1. We present a meta-analysis of neural topic model evaluation (Section 3);
2. we develop standardized, pre-processed versions of two widely-used English-language evaluation datasets, along with a transparent end-to-end code pipeline for reproduction of results (Section 4.1)[5];
3. we optimize three topic models—one classical and two neural—using identical preprocessing, model selection criteria, and hyperparameter tuning (Section 4.2);
4. we evaluate these models using human ratings and word intrusion tasks (Section 5); and
5. we provide new evaluations of the correlation between automated and human evaluations (Section 6).

Our findings challenge the validity of fully-automated evaluations as currently practiced: automated evaluation declares winners between models when the corresponding human evaluations cannot.

---

[3]Outside of the core method-development literature, human evaluations have been used to develop new metrics and improve understanding of existing model behavior (Bhatia et al., 2017; Morstatter and Liu, 2018; Lund et al., 2019; Alokaili et al., 2019, *inter alia*).

[4]We use "classical" to mean generative models defined by a chain of conjugate exponential family distributions optimized by Gibbs sampling or variational inference.

[5]`github.com/ahoho/topics`

## 2 Operationalizing Topic Coherence

A topic model is a probabilistic generative model of text that uses latent *topics* to summarize a larger collection of documents. The most influential variant, latent Dirichlet allocation (Blei et al., 2003, LDA), assumes that $K$ latent topics are distributions over word types, $\beta_k$, and that the documents $\mathcal{D}$ are admixtures over the topics, $\theta_d$. Users often evaluate model outputs globally, focusing on the most probable $N$ words of each topic, and locally, considering the most probable topics for each document.

While techniques for topic modeling have progressed from variational inference (Blei et al., 2003) to Gibbs sampling (Griffiths and Steyvers, 2004) to deep generative approaches (Srivastava and Sutton, 2017; Wang et al., 2020b), the core goal discussed in Section 1, obtaining human-understandable categories, remains central. The latest wave of methods, *neural topic models* (NTM), use continuous word representations and gradient optimization to fit parameters. These models claim to produce more interpretable topics than other prior methods, including LDA.

Those claims are supported by improvements on automated measures of topic coherence.

### 2.1 Human Metrics of Topic Coherence

Like the concept of *interpretability*, that of real-world *coherence* is "simultaneously important and slippery" (Lipton, 2018). We will not attempt to formalize it here—though see discussion in Section 7. For present purposes, the term has its roots in Latin *cohaerere*, "to stick together," and we will think of coherence as an intangible sense, available to human readers, that a set of terms, when viewed together, enable human recognition of an identifiable category.[6] We review two human ratings of topic quality: direct ratings and intrusion.

**Rating**    Raters see a topic and then give the topic a quality score, conventionally on a three-point ordinal scale (Newman et al., 2010; Mimno et al., 2011; Aletras and Stevenson, 2013, *inter alia*).

**Intrusion**    Chang et al. (2009) devise the *word intrusion* task as a behavioral way to assess topic coherence. The core idea is that when the top words in a topic identify a coherent latent category, it is easier to identify words that do not belong to that category. Operationally, each topic is represented as its top words plus one "intruder" word which has a low probability of belonging to that topic, but a high probability of belonging to a different topic. Topic coherence is then judged by how well human annotators detect the "intruder" word.

### 2.2 NPMI: The Standard Automated Topic Model Coherence Evaluation

Using the word intrusion task, Chang et al. (2009) showed that perplexity—the original topic model evaluation metric—*negatively* correlates with human evaluations of topic quality. This finding revealed a need for an automated measurement of topic coherence: an automated metric can measure model quality without expensive, time-consuming, and difficult-to-reproduce human experiments.

Lau et al. (2014) find some metrics that *positively* correlate with human intrusion and rating scores, particularly when aggregating scores over all topics from a given model. Because of that validation, the prevailing evaluation for model comparison is pairwise normalized pointwise mutual information. NPMI scores topics highly if the top $N$ words—summed over all pairs $w_i$ and $w_j$—have high joint probability $P(w_j, w_i)$ compared to their marginal probability:[7]

$$\sum_{j=2}^{N} \sum_{i=1}^{j-1} \frac{log \frac{P(w_j, w_i)}{P(w_i) P(w_j)}}{-log P(w_i, w_j)}. \tag{1}$$

The probabilities are estimated using word co-occurrence counts from a *reference corpus* for a specific context window (which can range from ten words to the entire document). As a result, the choice of reference corpus determines the strength of human correlation (Lau et al., 2014; Röder et al., 2015).

---

[6]This perspective aligns with Propositions 2 and 3 of Doogan and Buntine (2021): "an interpretable topic is one that can be easily labeled," and "has high agreement on labels."

[7]Alternative metrics exist, but they typically also rely on either joint probability estimates or NPMI directly (e.g., $C_v$ Röder et al., 2015).

| Evaluation | Count | | Experimentation | Count | |
|---|---|---|---|---|---|
| Number of human evaluations | 0 | (0%) | *Preprocessing* | | |
| *Automated Coherence* | | | Inconsistent over datasets | 12 | (30%) |
| Metric | | | Ambiguous preprocessing | 9 | (23%) |
| NPMI | 26 | (72%) | *Model comparisons* | | |
| Other | 22 | (61%) | All models tuned | 5 | (13%) |
| Explicit implementation | 22 | (61%) | Unclear h.param search | 16 | (40%) |
| Explicit ref. corpus | 10 | (28%) | Unclear LDA baseline, if used | 7 | (24%) |
| Perplexity w/o coherence | 3 | (8%) | Recent baseline (w/in 2 yrs) | 31 | (78%) |
| | | | Multiple runs / sig. testing | 11 | (28%) |

Table 2: Meta-analysis of forty neural topic modeling papers (denominator may change, as not all conditions are applicable). No recent neural topic modeling papers use human evaluations of coherence, and the metrics and models are difficult to replicate.

A measurement is *valid* to the extent that it measures what it is intended to measure in the real world. Historically, automated coherence has been validated using *human* judgements from either crowdworkers (Newman et al., 2010; Aletras and Stevenson, 2013) or experts (Mimno et al., 2011). However, correlations based on classical models may not be applicable for NTMs. Our skepticism is motivated by theory, as neural word representations are intimately connected to NPMI, as explicitly used by Aletras and Stevenson (2013) and which produce similar NPMI scores as Lau et al. (2014). Levy and Goldberg (2014) show that multiple representations create factorizations of PMI matrices. Topic models that have access to these rich representations (e.g. Dieng et al., 2020, and others) could thus create topics with good NPMI scores without explaining the corpus well to a user. In contrast to classical topic models, no one has investigated the validity of NPMI evaluation for NTMs.

Given this lacuna, we conduct experiments aimed at validating that automated topic evaluations still correlate with human judgments of neural topic model quality. We compare against two common human evaluations of individual topic quality: direct rating and intrusion. Human evaluations, like automated topic modeling, lack standardization, which we address in Section 5.

## 3 A Meta-Analysis of Neural Topic Modeling

We survey the neural topic modeling (NTM) literature to assess the state of evaluation in contemporary topic model development. First, we take all references made by an existing, comprehensive survey of NTMs (Zhao et al., 2021b), from which we select (a) modeling papers which (b) mention topic interpretability and (c) compare models' topics with an existing baseline. This yields forty models, which all claim superior topic coherence. We examine data processing steps, hyperparameter tuning, baseline selection, and automated coherence calculations. Table 2 summarizes our results and Appendix A.1 enumerates the papers.

Our analysis reveals variance in all areas. Preprocessing, which can significantly affect model quality and automated metrics, is often (30%) inconsistent across datasets within the same paper. When preprocessing *is* consistent, authors omit details necessary to fully replicate the pipeline. These issues imply that automated metrics for the same baselines and source datasets vary across papers. Compounding the problem, researchers often train their models on different datasets from those used to establish the relationships between human annotations and automated metrics; Doogan and Buntine (2021) find that the same metrics may not predict interpretability in new domains. Mirroring findings from Dodge et al. (2019), 40% of papers fail to clearly specify their model tuning procedure, often even the metric used for model selection.

Calculation of automated coherence metrics is equally fraught. As discussed in Section 2.2, a complete specification for NPMI involves several pieces of information, including the reference corpus used to estimate joint word probabilities, the co-occurrence window size, and the number of words selected from the head of the topic distribution. Three out of four papers fail to explicitly indicate the reference corpus; even when we can assume the input corpus is used (13 cases), it remains uncertain whether authors use, e.g., a held-out set or the training documents themselves. For the 61% that specify the implementation of their coherence metric (by pointing to a code repository or writing out the formula), some of these factors may still be in question. For instance, six authors

reference Lau et al. (2014) and the supporting code,[8] but the implications are ambiguous: the original paper suggests a large corpus from the same source as the training data, but the repository script defaults to Wikipedia. In other cases, authors use bespoke implementations, which creates room for errors, or deviate from the settings used in human experiments. For example, several papers use a document-wide context window with NPMI, which has not been correlated with human judgments.

Last, *even if* automated evaluations are consistent, all claims of coherence improvement depend on the validity results in Lau et al. (2014) generalizing to neural topic models.

## 4 Closing the Standardization Gap for Topic Models

Our human evaluation of topic model outputs serves multiple purposes: (a) establishing whether NTMs show improved coherence over a classical baseline and (b) re-evaluating the efficacy and reliability of automated coherence metrics. In addition, a key goal is (c) to provide a standardized preprocessing pipeline to support head-to-head comparisons as new methods are developed.[9]

We identify two commonly-used datasets, which we in turn process using a standard pipeline. We then estimate topic models on each dataset following a computationally fair hyperparameter search. Our standardization efforts are similar to concurrent work by Terragni et al. (2021); the main differences are that we (a) mandate consistent preprocessing between training and reference corpora, (b) support multi-word expressions during vocabulary creation (see below), and (c) support distributed hyperparameter searches.

### 4.1 Datasets and Preprocessing

Following Chang et al. (2009), we use English articles from Wikipedia and the New York *Times* (Table 7). For Wikipedia, we use Wikitext-103 (WIKI, Merity et al., 2017), and for the *Times*, we subsample roughly 15% of documents from LDC2008T19 (NYT, Sandhaus, 2008), making it an order of magnitude larger than WIKI. To compute reference counts, we use a 4.6M document Wikipedia dump from September 2017 and the full 1.8M document LDC2008T19 set, processed identically to the training data.

We use SpaCy (Honnibal et al., 2020) to tokenize and identify entities in the text. We create new tokens for detected entities of the form `New_York_City`, per Krasnashchok and Jouili (2018). Schofield and Mimno (2016) find that lemmatization and word-stemming can hurt English topic interpretability, so we do not lemmatize. To maintain a roughly equal vocabulary size over datasets, we use a power-law relationship of corpus size (c.f. Zipf, 1949) to rule out tokens occurring in fewer than a given number of documents.[10] In addition to a standard stopword list, we define corpus-specific stopwords as tokens appearing in more than 90% of documents. See Appendix A.2 for complete preprocessing details.

### 4.2 Models

We evaluate one venerable classical model and two newer neural models:

**Gibbs-LDA**   As a strong classical baseline, we use the widely-loved Mallet (McCallum, 2002) implementation of Gibbs-sampling for LDA (Griffiths and Steyvers, 2004). Mallet produces topics of (qualitatively) competitive quality to neural models (Srivastava and Sutton, 2017).

**Dirichlet-VAE**   We reimplement Dirichlet-VAE (Burkhardt and Kramer, 2019), a state-of-the-art NTM. For simplicitly, we use pathwise gradients for the Dirichlet (Jankowiak and Obermeyer, 2018), rather than the rejection sampling variational inference of the authors' primary variant.[11] Dirichlet-VAE is a wholesale improvement on one of the first successful NTMs, the popular ProdLDA (Srivastava and Sutton, 2017), and is competitive against recent models on automated coherence. The generative

---

[8]`github.com/jhlau/topic_interpretability`

[9]Our preprocessing pipeline is agnostic to dataset and easily portable. `github.com/ahoho/topics`

[10]We target vocabularies approximating the number of words known by an adult English-speaker (Brysbaert et al., 2016): roughly 40k for WIKI and 35k for NYT.

[11]We replicate their NPMI and redundancy scores on 20 newsgroups. `github.com/ahoho/dvae`

Select which term is the least related to all other terms and your familiarity with the words

Terms
- ○ painting
- ○ paintings
- ○ casualties
- ○ painter
- ○ literary
- ○ poems

Answer Confidence
- ○ I am familiar with most of these terms.
- ○ I am **not** familiar with most of these terms, but I **can** answer confidently.
- ○ I am **not** familiar with most of these terms, and so I **cannot** answer confidently.

Figure 1: The word intrusion task presented to crowdworkers (the ratings task is in Appendix A.4).

model is simple and retains a broad similarity to LDA. The primary difference is that it does not constrain the estimated topic-word distributions to the simplex.

**ETM**   Thanks to their improved flexiblity, many NTMs incorporate external word representations, on the premise that large-scale, general language knowledge improves topic quality (Bianchi et al., 2021; Hoyle et al., 2020). The Embedded Topic Model (Dieng et al., 2020) is a popular NTM that relies on word embeddings in its generative model.[12]

We maintain a fixed computational budget per model following the exhortation of Dodge et al. (2019) and use a random set of 164 hyperparameter settings across datasets for each model type.[13] We train models for a variable number of steps (a hyperparameter); to calculate automated coherence for the model, we use the topics produced at the last step. For human evaluations, we select the models that maximize NPMI, estimated using the reference corpus with a ten-word window over the top ten topic words, per Lau et al. (2014). We follow the recommendation of Dieng et al. (2020) and learn skip-gram embeddings on the training corpus for ETM (experiments with external pretrained embeddings did not yield substantially different results). As in Hoyle et al. (2020), we eliminate models with highly redundant topics, a known degeneracy of NTMs (Burkhardt and Kramer, 2019): (a) models in which any of the top five words of one topic overlap with another and (b) models that have a topic uniqueness score (Nan et al., 2019) above 0.7. Ranges for hyperparameters and other details are in Appendix A.3.

## 5   Human Evaluations of Topic Quality

We use the *ratings* and *word intrusion* tasks from Section 2.2 as human evaluations of topic quality. We recruit crowdworkers using Prolific.co, an online panel provider and collect data with the Qualtrics survey platform. We pay workers 2.5 USD per ratings survey and 3 USD per word intrusion survey, equivalent to 15 USD/hour.

In order to draw meaningful conclusions from human annotations, we require an adequate number of participants to ensure acceptable statistical power. However, Card et al. (2020) show that many NLP experiments, including those relying on human evaluation, are insufficiently powered to detect model differences at reported levels. Adopting a straightforward generative model of annotations (Appendix A.5), we select enough crowdworkers per task to ensure sufficient statistical power (at least $1 - \beta = 0.9$) to obtain significance at $\alpha = 0.05$, resulting in a minimum of fifteen crowdworkers per topic for both tasks. On this criterion, both Chang et al. (2009) and thus Lau et al. (2014), with eight annotators, are underpowered.

For each of our two datasets, we generate fifty topics each from the three models in Section 4.2. In the word intrusion task, we sample five of the top ten topic words plus one intruder; for the ratings task, we present the top ten words in order (Figure 4). We separate the datasets for each task and

---

[12]github.com/adjidieng/ETM

[13]While runtimes can vary drastically by model, this study is not concerned with implementation efficiency (although efficiency matters, see Ethayarajh and Jurafsky, 2020).

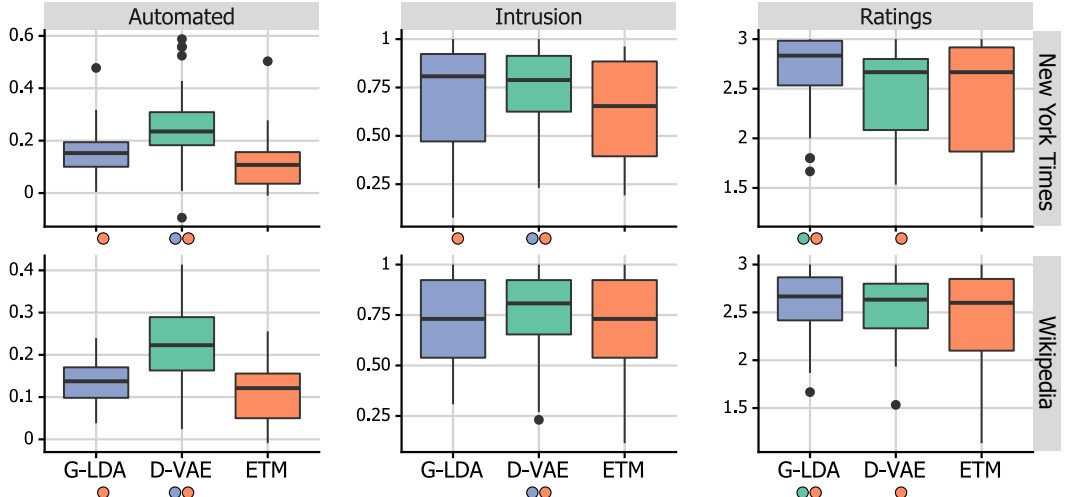

Figure 2: While automated evaluations (here, NPMI) suggest a clear winner between models, human evaluation is more nuanced. Human judgments exhibit greater variability over a smaller range of values. Colored circles correspond to pairwise one-tailed significance tests between model scores at $\alpha = 0.05$; for example, the rightmost orange circle at bottom right shows that human intrusion ratings for D-VAE are significantly higher than ETM for topics derived from Wikipedia.

randomly sample 40 of the 150 topics. In the ratings task, we include an additional sixteen synthetic poor-quality topics to help calibrate scores and filter out low-quality respondents.[14]

Phrasing of questions closely follows the wording used by Chang et al. (2009), and crowdworkers received detailed instructions with examples (Appendix A.4) before responding to items.[15] As topics can be esoteric (e.g., last columns of Table 1), we ask crowdworkers about their familiarity with the words in each question. We speculate that this question can help protect against spurious low scores for otherwise coherent topics, as real-world users of topic models are usually familiar with domain-specific terminology (see further discussion in Section 7).

# 6   Human Judgment Differs From Automated Metrics

We compare human judgments to automated methods on topics estimated using our three models.

## 6.1   Human Assessment

To establish model differences using human ratings, we use pairwise significance tests: a proportion test for the intrusion scores, a $U$ test (Mann and Whitney, 1947) for the ratings, and a $t$-test for automated metrics (Figure 2), using one-tailed tests for each pair in both directions. Although D-VAE fares better on the intrusion task, evaluation using ratings favors G-LDA.[16]

Our human evaluation results are consistent with past iterations of the ratings and word intrusion tasks for topic models. Mimno et al. (2011) report an average of 2.36 on the ratings task on a dataset of medical paper abstracts.[17] Our ratings means are 2.5 to 2.8 across all variations (Figure 2). Our word intrusion means range from 0.7 to 0.8, which is comparable to the roughly 0.8 accuracy on the LDA model evaluated in Chang et al. (2009). Median time taken on the tasks was 8–9 minutes.

---

[14]For generating *synthetic poor-quality topics*, we use random high-probability words appearing in topics from other hyperparameter settings, but that have low probability among selected topics. Eight topics each are generated from the vocabularies of NYT and WIKI.

[15]Code to convert topic model output into deployable questionnaires is at github.com/ahoho/topics.

[16]These discrepancies among human tasks support the argument that standard coherence metrics alone may be insufficient for automated model selection (Doogan and Buntine, 2021).

[17]Newman et al. (2010) and Lau et al. (2014) do not report an average.

|  | Ref. Corpus → Train Corpus ↓ | NPMI (10-token window) | | | | $C_v$ (110-token window) | | | |
|---|---|---|---|---|---|---|---|---|---|
|  |  | NYT | WIKI | Train | Val | NYT | WIKI | Train | Val |
| Intrusion | NYT | 0.27 | 0.43 | 0.27 | 0.24 | 0.34 | **0.45** | 0.35 | 0.34 |
|  | WIKI | 0.34 | 0.36 | **0.39** | 0.17 | 0.32 | 0.34 | 0.34 | 0.20 |
|  | Concatenated | 0.29 | 0.40 | 0.32 | 0.17 | 0.32 | **0.40** | 0.35 | 0.24 |
| Rating | NYT | 0.37 | **0.48** | 0.37 | 0.39 | 0.41 | 0.46 | 0.44 | 0.45 |
|  | WIKI | 0.34 | 0.41 | **0.44** | 0.28 | 0.32 | 0.40 | 0.40 | 0.34 |
|  | Concatenated | 0.37 | **0.44** | 0.41 | 0.35 | 0.38 | 0.42 | 0.42 | 0.42 |

Table 3: Spearman correlation coefficients between mean human scores and automated metrics. Underlined values have overlapping bootstrapped 95% confidence intervals with that of the **largest** value in each row. "Concatenated" refers to correlations computed on a concatenation of values for the NYT and WIKI items. "Val" is a small held-out set of 15% of the training corpus. Using the more data-appropriate logistic and ordered probit regressions for word intrusion and ratings data leads to different conclusions about relative metric strength (Appendix Table 10). CIs are estimated using 1,000 samples.

Following Aletras and Stevenson (2013), we calculate inter-annotator agreement with the mean Spearman correlation between each respondent's score per topic and the average of other respondent scores, obtaining a value of 0.75 (compare to their value of 0.7 on the NYT corpus). Additionally, we include synthetic poor-quality topics (footnote 14)—correctly identified by annotators—and we monitor the duration taken for the survey to hedge against insincere submissions.

## 6.2 Automated Metrics

NPMI declares D-VAE the unequivocal victor among the three models (with G-LDA a clear second), a very different story from the human judgments. To understand the relationship between automated metrics and human ratings, we estimate the Spearman correlation between the two sets of values for each task and dataset for metric variants (Table 3). Although previous studies have used mean human ratings over topics, this decision obscures the inherent variance of the human ratings and leads to overconfident estimates. We therefore construct 95% confidence intervals by resampling ratings, with replacement, equal to the number of annotators per task (Table 3). We estimate NPMI with the standard 10-word window and $C_v$ (Röder et al., 2015) with the recommended 110-word window.[18] The Wikipedia corpus appears to be best correlated with human judgments, even for the models trained on the NYT corpus—this contradicts Lau et al. (2014), where within-domain data have the highest correlations.

While all correlation coefficients are statistically significant, the strength of the correlation alone does not justify their use in model selection, as is standard in the NTM literature (Section 3). In particular, the inherent uncertainty of human judgments means that it is difficult to determine when an increase in a model's mean automated coherence implies a significant improvement in the corresponding human scores.[19]

As noted above (Figure 2), automated metrics exaggerate model differences compared to human judgments. To help clarify the utility of automated metrics for model selection, we ask how often an automated metric incorrectly asserts that one model is superior to another. To do so, we generate a bootstrapped estimate of the false discovery rate of each model. First, for each dataset, we randomly sample two independent sets of $K = 50$ topics (without replacement) from the original pool of 150, along with their corresponding automated and human scores (resampled with replacement, as in Table 3). Treating the two sampled sets as outputs from two different models, we compute pairwise significance tests between each set for both the $K$ automated metrics and $K \times M$ human scores (using a proportions $z$-test for the intrusion scores and $t$-tests for all other values). After repeating this process for $N = 1000$ iterations, we report the proportion of significant differences detected using

---

[18]We use gensim (Řehůřek and Sojka, 2010) to calculate coherence. We process the reference corpora identically to the training data, retaining only terms that exist in the training vocabulary. Other metrics, like $C_{\text{UCI}}$ (Newman et al., 2010) and $C_{\text{UMASS}}$ (Mimno et al., 2011), show low correlations.

[19]Better models of human scores could help quantify this relationship (e.g., GLMs, see Appendix A. 10).

| | Ref. Corpus → Train Corpus ↓ | NPMI (10-token window) | | | $C_v$ (110-token window) | | |
|---|---|---|---|---|---|---|---|
| | | NYT | WIKI | Train | NYT | WIKI | Train |
| Intrusion | NYT | 46 / 53 | 34 / 48 | 48 / 50 | 35 / 38 | **30** / 29 | 34 / 35 |
| | WIKI | 44 / 76 | 33 / 78 | 33 / 75 | 45 / 48 | 38 / 49 | **37** / 45 |
| | Concatenated | 42 / 67 | 40 / 66 | 41 / 64 | 36 / 46 | **31** / 44 | 30 / 45 |
| Rating | NYT | 45 / 50 | 45 / 51 | 41 / 47 | 27 / 29 | 26 / 26 | **21** / 26 |
| | WIKI | 40 / 73 | 31 / 73 | 33 / 71 | 38 / 40 | 31 / 40 | **28** / 34 |
| | Concatenated | 39 / 66 | 36 / 66 | 37 / 62 | 31 / 38 | 28 / 38 | **19** / 36 |

Table 4: False discovery rate (1−precision, lower is better) and false omission rate of significant model differences when using automated metrics; automated metrics often overstate meaningful model differences. **Bolded** values are those with the lowest geometric mean of FDR and FOR. We sample two independent sets of 50 topics along with their human scores and automated metrics; these sets act as the outputs of two "models". We then compute significance tests between sets (per Figure 2) on both the automated scores and human scores. A false positive occurs when one set has significantly larger automated scores despite no meaningful difference in actual human scores. Estimates are over 1,000 samples.

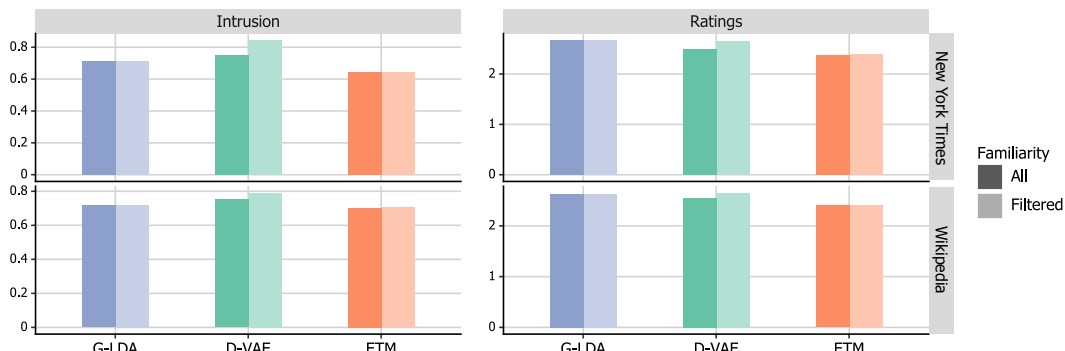

Figure 3: Mean human evaluation on the ratings and word intrusion tasks, after filtering out respondents who reported a lack of familiarity with the topic words. When filtering, D-VAE scores improve, highlighting its tendency to produce esoteric topics.

the predicted scores despite *equivalent* human scores (after correcting for the probability of type I errors, $\alpha = 0.05$).[20] Even the best-performing automated metrics predict significant differences absent a meaningful human effect roughly one-fifth of the time (Table 4).

These results suggest that automated metrics alone may be inadequate for model comparison.

### 6.3 Explaining the discrepancy

One reason for the discrepancy between human judgments and automated metrics is that metrics favor more esoteric topics. Specifically, there is a significant negative correlation between a topic's NPMI or $C_v$ and the share of respondents reporting familiarity with topic words (Pearson's $\rho = -0.29$). And while D-VAE achieves the highest automated metric scores of the three models, it produces topics with the fewest familiar words: respondents report familiarity with terms over 90% of the time on both tasks for G-LDA and ETM, but they do so only 70% of the time for D-VAE. This difference suggests that the topics selected by D-VAE are narrower in scope than those of the other models. As shown in Figure 3, removing item annotations where respondents indicate unfamiliarity causes both accuracy in the word intrusion task and the ratio of "Very related" terms in the ratings task for D-VAE to increase substantially.

Qualitatively, this result is apparent when examining topics with a high NPMI but low humans ratings. In Table 5, the top rows consists of financial terms that frequently appear *together* in NYT articles,

---

[20]Details on testing equivalence are in Section A.5.1.

| Data | Model | Topic | NPMI | Rat. | Int. |
|---|---|---|---|---|---|
| NYT | D-VAE | inc 6mo earns otc rev qtr 9mo nyse outst dec | 0.56 | 1.60 | 0.77 |
| WIKI | D-VAE | waterline conning turrets boilers amidships aft knots armament guns mounts | 0.33 | 1.93 | 0.65 |
| NYT | G-LDA | bedroom room bath taxes year market listed kitchen broker weeks | 0.30 | 2.00 | 0.23 |
| NYT | D-VAE | condolences mourns mourn board_of_directors heartfelt deepest esteemed | 0.38 | 2.60 | 0.23 |
| NYT | D-VAE | shareholders earnings federated mci shares takeover new_york_stock_exchange | 0.18 | 3.00 | 0.81 |
| WIKI | D-VAE | continental_army expedition militia frigate musket frigates muskets skirmish | 0.11 | 3.00 | 0.69 |
| NYT | D-VAE | medicaid medicare hospitals welfare uninsured patients | 0.13 | 2.80 | 0.96 |
| NYT | G-LDA | city mayor state new_york new_york_city officials county yesterday governor | 0.09 | 2.53 | 1.00 |

Table 5: Topics with the largest human–NPMI discrepancies; top half are topics where NPMI is high and human preferences are low, bottom half is the reverse. NPMI favors esoteric and corpus-specific topics. **NPMI** is calculated with a 10-token sliding window over the in-domain reference corpus, **Rat.** is the average 3-point rating for a topic, and **Int.** refers to the percentage of annotators who identify the intruder word.

and the second row contains rare terms about boating—arguably both are reasonable topics for their respective corpora. We can also see instances where words are qualitatively very related (bottom half of table), but that NPMI fails to score high—perhaps because these words, while related, may not frequently appear together within a ten-word sliding window (Equation 1).

Even for familiar words, some topics may be sensible in the context of the specific corpus, despite their component words lacking an immediately obvious semantic relationship. For example, the topic words in the third and fourth rows appear somewhat unrelated (e.g., "taxes" and "bedroom" in the third row), but they are in fact characteristic of common document types in the New York *Times*: real estate listings and obituaries. Topics like these render the word intrusion task more difficult: only 23% of crowdworkers identified the intruder for both topics.

Furthermore, using term familiarity as a proxy for domain expertise does not address the key problems with topic model evaluation: even after filtering out respondents who are not familiar with topic terms, automated metrics still overstate model differences (Appendix A.7). The problems with topic model evaluation may therefore extend to our choice of *human* evaluations as well.

## 7 So... is Automated Topic Modeling Evaluation Broken?

To the extent that our experimentation accurately represents current practice, our results do suggest that topic model evaluation—both automated and human—is overdue for a careful reconsideration. In this, we agree with Doogan and Buntine (2021), who write that "coherence measures designed for older models [. . . ] may be incompatible with newer models" and instead argue for evaluation paradigms centered on corpus exploration and labeling. The right starting point for this reassessment is the recognition that both automated and human evaluations are abstractions of a real-world problem. The familiar use of precision-at-10 in information retrieval, for example, corresponds to a user who is only willing to consider the top ten retrieved documents. In future work, we intend to explore automated metrics that better approximate the preferences of real-world topic model users.

One primary use of topic models is in computer-assisted content analysis. In that context, rather than taking a methods-driven approach to evaluation, it would make sense to take a needs-driven approach.[21] Generic evaluation of topic models using domain-general corpora like NYT needs to be revisited, since there is no such thing as a "generic" corpus for content analysis, nor a generic analyst. *Content analysis* can be formulated in a broad way, as Krippendorff (2004) has shown, but its actual application is always in a domain, by people familiar with that domain. This fact stands in tension with the desirable practicalities of general corpora and crowdworker annotation, and the field will need to address this tension. We have identified "coherence" as calling out a latent concept in the mind of a reader. It follows that we must think about who the relevant human readers are and the conceptual spaces that matter to them.

---

[21]These needs also have a computational component: neural models usually have longer runtimes even when accelerated with GPUs, whereas many practitioners work in local, CPU-only, environments. See Appendix A.3 for additional details on runtimes.

## Acknowledgements

This material is based upon work supported by the National Science Foundation under Grants 2031736, 2008761, 1822494, ARLIS, and by an Amazon Research Award. We thank Sweta Agrawal for her suggestion to conduct a meta-analysis. We owe much appreciation to Dallas Card for his keen advice on power analyses. Thanks to Frank Fineis for help on several statistical questions, as well as Shuo Chen for his suggestions regarding the false discovery rate calculations. Finally, we thank Caitie Doogan for her helpful comments on the clarity of argumentation, as well as our anonymous reviewers.

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
