# A  Appendix

## A.1  List of Neural Topic Modeling Works used in our Meta-Analysis

In Table 6, we report the forty publications used in our meta-analysis (Section 3), which are sourced from a survey of neural topic models (Zhao et al., 2021b).

## A.2  Preprocessing Details

Our steps are delineated in our implementation,[22] but we list our choices here for easy reference. Corpus statistics are in Table 7. We use the default `en-core-web-sm` spaCy model (Honnibal et al., 2020), version 3.0.5, throughout.

Document processing
- We do not process documents with fewer than 25 whitespace-separated tokens.
- Following processing (e.g., stopword removal), we remove documents with fewer than five tokens.
- We truncate documents to 5,000 whitespace-separated tokens for NYT and to 19,000 for WIKI (in both cases affecting less than 0.15% of documents).

Vocabulary creation
- We tokenize using spaCy.
- We lowercase terms.
- We *do not* lemmatize.
- We detect noun entities with spaCy, keeping only the ORG, PERSON, FACILITY, GPE, and LOC types, joining constituent tokens with an underscore (e.g, "New York City" → `new_york_city`).

Vocabulary filtering
- The vocabulary is created from the training data. The reference texts used in coherence calculations are processed identically and use the same vocabulary.
- We filter out stopwords using the default spaCy English stopword list.[23] Stopwords are retained if they are contained within detected noun entities (e.g., "The United States of America" → `united_states_of_america`).
- We filter out tokens with two or fewer characters.
- We retain only tokens that are matched by the regular expression `^[\w-]*[a-zA-Z][\w-]*$`
- We remove tokens that appear in more than 90% of documents.
- We remove tokens that appear in fewer than $2(0.02|D|)^{1/\log 10}$ documents, where $|D|$ is the corpus size.[24]

## A.3  Training Details

Expanding Section 4.2, we detail the hyperparameter tuning for each of our three topic models, along with other pertinent details about runtimes and compute resources. Scripts used to run the models with all the various hyperparameter configurations are released as part of our code; this section is also included for reference.

Our general strategy, especially with the neural models, is to select different values around the reported optimal settings in original papers. For all three models, we try two different values for the number of training iterations (G-LDA) or epochs (D-VAE, ETM).

---

[22]`github.com/ahoho/topics`

[23]`github.com/explosion/spaCy/blob/v3.0.5/spacy/lang/en/stop_words.py`

[24]Standard rules-of-thumb for vocabulary pruning, like removing terms that appear in fewer than 0.5% of documents (Denny and Spirling, 2018), ignore the power-law distribution of word frequency Zipf (1949), and hence do not scale to large corpora. To keep vocabulary sizes roughly consistent across datasets, we set the minimum document-frequency for terms as a (power) function of the total corpus size. This has the intuitive appeal of increasing proportional to the order of magnitude of the number of total documents, starting at a minimum document-frequency of 2 for a 50-document corpus and reaching about 110 for a corpus of 500,000.

| Source | Human Evals? | Perplexity | Coherence | Implementation Specified | Ref. Corpus Specified ? | Consistent Preproc? | Hparam search? | >1 run / err. bars? | LDA Implementation? | Baseline w/in 2 yr? |
|---|---|---|---|---|---|---|---|---|---|---|
| Bianchi et al. (2021) | No | No | NPMI, Embed-sim | None | Internal, External-GoogleNews | Yes | No | Yes | Variational | No |
| Zhao et al. (2021a) | No | No | NPMI | Palmetto | No | Unclear | No | Yes | N/A | Yes |
| Feng et al. (2020) | No | Yes | NPMI | None | No | Yes | No | No | N/A | Yes |
| Hoyle et al. (2020) | No | No | NPMI | In paper | External NYT, Internal | No | Yes | Yes | N/A | Yes |
| Hu et al. (2020) | No | No | $C_p$, $C_a$, NPMI | Palmetto | External WIK1 | No | Likely no | No | Sampling | Yes |
| Isonuma et al. (2020) | No | Yes | NPMI | None | No, likely external | Unclear | No | No | Sampling | No |
| Joo et al. (2020) | No | Yes | NPMI | None | No, likely internal | No | Likely yes | No | N/A | Yes |
| Lin et al. (2020) | No | Yes | NPMI | None | No, likely internal | Unclear | Yes | Yes | N/A | Yes |
| Ning et al. (2020) | No | Yes | NPMI | Lau github | No | Yes | Likely no | Yes | Variational | No |
| Panwar et al. (2020) | No | No | NPMI | Lau github | No | Yes | Likely no | No | Sampling | Yes |
| Rezaee and Ferraro (2020) | No | No | N/A | N/A | N/A | Yes | Likely no | Yes | Variational | No |
| Thompson and Mimno (2020) | No | No | Coherence, PMI | In paper | External NYT | No | No | Yes | Sampling | No |
| Tian et al. (2020) | No | Yes | NPMI | None | No | No | Yes | No | Variational | Yes |
| Wang et al. (2020a) | No | No | $C_p$, $C_a$, NPMI, UCI | Palmetto | No | No | No | No | Sampling | Yes |
| Wu et al. (2020a) | No | Yes | NPMI | None | No | No | Yes | No | N/A | Yes |
| Wu et al. (2020b) | No | No | $C_v$ | Palmetto | No | Yes | No | No | Unspecified | Yes |
| Yang et al. (2020) | No | Yes | Coherence | In paper | No, likely internal | Yes | No | No | Unspecified | No |
| Zhou et al. (2020) | No | No | NPMI, $C_p$ | Palmetto | External WIK1 | No | Likely no | No | Unspecified | Yes |
| Burkhardt and Kramer (2019) | No | Yes | NPMI | None | No, likely internal | Unclear | Yes | No | Variational | Yes |
| Dieng et al. (2020) | No | Yes | Coherence | In paper | No, likely internal | Yes | No | No | Unspecified | No |
| Gui et al. (2019) | No | No | $C_v$ | None | External WIK1 | Yes | Likely no | No | Unspecified | Yes |
| Gupta et al. (2019b) | No | Yes | $C_v$ | Gensim | No, likely internal | Unclear | Likely no | No | N/A | No |
| Gupta et al. (2019a) | No | Yes | $C_v$ | Gensim | No, likely internal | Unclear | Likely no | No | Sampling | Yes |
| Lin et al. (2019) | No | Yes | PMI | In paper | No, likely internal | Unclear | No | No | Variational | Yes |
| Liu et al. (2019) | No | Yes | NPMI | Lau github | No, likely internal | Yes | No | No | Variational | Yes |
| Nan et al. (2019) | No | No | NPMI | None | No | No | No | No | Sampling | Yes |
| Wang et al. (2020b) | No | No | $C_p$, $C_a$, UCI, NPMI, UMASS | Palmetto | No | Yes | No | No | Unspecified | Yes |
| Card et al. (2018) | No | Yes | NPMI | In paper | External-gigaword | Yes | Likely yes | No | Sampling | Yes |
| Ding et al. (2018) | No | Yes | NPMI | Lau github | No, likely external | No | Likely no | No | Sampling | Yes |
| He et al. (2018) | No | No | Coherence | None | No, likely internal | Yes | No | No | N/A | Yes |
| Peng et al. (2018) | No | Yes | N/A | N/A | N/A | Yes | Likely no | No | Variational | Yes |
| Silveira et al. (2018) | No | Yes | NPMI | Lau github | Internal | Yes | No | Yes | N/A | Yes |
| Zhang et al. (2018) | No | Yes | N/A | N/A | N/A | Unclear | Likely no | No | N/A | Yes |
| Zhao et al. (2018) | No | Yes | NPMI | Palmetto | External WIK1 | Unclear | No | Yes | N/A | Yes |
| Zhu et al. (2018) | No | No | Coherence | None | No, likely internal | Yes | Likely no | No | Variational | Yes |
| Jung and Choi (2017) | No | Yes | NPMI, PMI, UMASS | In paper | No | No | No | No | Sampling | Yes |
| Miao et al. (2017) | No | Yes | NPMI | None | No | Yes | Likely no | No | Variational | Yes |
| Srivastava and Sutton (2017) | No | Yes | NPMI | In paper | No | Yes | No | No | Sampling | Yes |
| Miao et al. (2016) | No | Yes | N/A | N/A | N/A | Yes | Likely no | No | Unspecified | Yes |
| Nguyen et al. (2015) | No | No | NPMI | Lau github | External WIK1 | Yes | No | Yes | Sampling | No |

Table 6: Papers used in meta-analysis, Section 3

|  | WIKI | NYT |
|---|---|---|
| Domain | Encyclopedia | News |
| *Number of Docs.* | | |
|    Training | 28.5k | 273.1k |
|    Reference | 4.62M | 1.82M |
| Mean Tokens / Doc. | 1291 | 281 |
| Vocab. Size | 39.7k | 34.6k |

Table 7: Corpus statistics. Datasets vary in domain, average document length, and total number of documents. WIKI is from Merity et al. (2017) and NYT is from Sandhaus (2008).

**G-LDA** We use gensim (Řehůřek and Sojka, 2010) as a Python wrapper for running Mallet. In Table 8a, we tune hyperparameters $\alpha$ (topic density parameter) and $\beta$ (word density parameter) which can be thought of as "smoothing parameters" that reserve some probability for the topics (words) unassigned to a document (topic) thus far. Mallet internally optimizes hyperparameters, and the Optimization Interval controls the frequency of hyperparameter updates, measured in training steps.

**D-VAE** Our reimplementation of Dirichlet-VAE (Burkhardt and Kramer, 2019) largely uses the same hyperparameters as reported in that work. As shown in Table 8b, we vary the prior for the Dirichlet distribution ($\alpha$), the learning rate ($\eta$), the $L_1$-regularization constant for the topic-word distribution ($\beta_{reg.}$, not in the original model but inspired by Eisenstein et al., 2011), the number of epochs to anneal the use of batch normalization in the decoder ($\gamma_{BN}$, comes from Card et al., 2018), and the number of epochs to anneal the KL-divergence term in the loss ($\gamma_{KL}$) (it needs to be introduced slowly in the loss function due to the component collapse problem in VAEs (Bowman et al., 2016)).

**ETM** Following Dieng et al. (2020), we learn skip-gram embeddings on the training corpus using the provided script, which relies on gensim. As shown in Table 8c, we vary the learning rate ($\eta$), the $L_2$ regularization constant for the Adam (Kingma and Ba, 2015) optimizer ($W_{decay}$), and a boolean indicator of whether to anneal the learning rate ($\gamma_\eta$). If annealing is allowed, the learning rate gets divided by 4.0 if the loss on the validation set does not improve for more than 10 epochs, per the default settings of the model (preliminary experiments showed that annealing did not attain higher NPMI).

The runtimes for each of the models on each dataset are in Table 9. We used AWS ParallelCluster to provide a cloud-computing computing cluster. Neural models ran on NVIDIA T4 GPUs using `g4dn.xlarge` instances with 16 GiB memory and 4 CPUs.[25] G-LDA (Mallet) ran on CPU only, with `m5d.2xlarge` instances (with 32 GiB memory, 8 CPUs).[26]

## A.4 Instructions for Crowdworkers

Recruiting participants on Prolific.co for a Qualtrics survey produced results with higher inter-worker agreement than Mechanical Turk, based on a pilot test. Using the Prolific.co platform, we recruited respondents that met the criteria of living in the United States and listing fluency in English. Each respondent was paid through Prolific upon completion of the survey, at a rate corresponding to $15 an hour. The total amount spent on conducting all the surveys, including our pilot test, was $2084.91. We used automated scripts to generate separate Qualtrics surveys for each task that contained the topics for evaluation, available in our released code. Each respondent was shown 25% of the questions in each survey; the question selection and answer display order was chosen randomly via the survey configuration on Qualtrics. Figures 1 and 4 depict our word intrusion and ratings tasks, respectively. Crowdworkers receive instructions explaining the task (Figure 5) and the dataset (Figure 6).

---

[25]https://aws.amazon.com/hpc/parallelcluster/
[26]See https://aws.amazon.com/ec2/instance-types/ for further details.

| | Model: **G-LDA** | | |
|---|---|---|---|
| $\alpha$ | $\beta$ | Optim. Interval | #Steps |
| $\{0.01, 0.05, 0.1, 0.25^\dagger, 1.0^*, 5.0\}$ | $\{0.01, 0.05^*, 0.1^\dagger\}$ | $\{0, 10^\dagger, 100, 500^*\}$ | $\{1000^\dagger, 2000^*\}$ |

(a) Hyperparameter ranges for G-LDA. $\alpha$ is the topic density parameter. $\beta$ is the word density parameter. Optim. Interval sets the number of iterations between Mallet's own internal hyperparameter updates. #Steps are training iterations.

| | | Model: **D-VAE** | | | |
|---|---|---|---|---|---|
| $\alpha$ | $\eta$ | $\beta_{reg.}$ | $\gamma_{BN}$ | $\gamma_{KL}$ | #Steps |
| $\{0.001, 0.01^{*\dagger}, 0.1\}$ | $\{0.001, 0.01^{*\dagger}\}$ | $\{0.0^*, 0.01, 0.1^\dagger, 1.0\}$ | $\{0, 1^*, 100, 200^\dagger\}$ | $\{100^*, 200^\dagger\}$ | $\{200, 500^{*\dagger}\}$ |

(b) Hyperparameter ranges for D-VAE. $\alpha$ is the Dirichlet prior. $\eta$ is the learning rate. $\beta_{reg.}$ is the $L_1$-regularization of the topic-word distribution. $\gamma_{BN}$ and $\gamma_{KL}$ are the number of epochs to anneal the batch normalization constant and KL divergence term in the loss, respectively. #Steps are training epochs.

| | Model: **ETM** | | |
|---|---|---|---|
| $\eta$ | $W_{decay}$ | $\gamma_\eta$ | #Steps |
| $\{0.001^*, 0.002, 0.01, 0.02^{*\dagger}\}$ | $\{1.2e^{-5*}, 1.2e^{-6\dagger}, 1.2e^{-7}\}$ | $\{0^{*\dagger}, 1\}$ | $\{500, 1000^{*\dagger}\}$ |

(c) Hyperparameter ranges for ETM. $\eta$ is the learning rate. $W_{decay}$ is the $L_2$ regularization constant. $\gamma_\eta$ is an indicator of whether learning rate is annealed. #Steps are training epochs.

Table 8: Hyperparameter settings for G-LDA, D-VAE, and ETM. $*$: Best setting for WIKI, $\dagger$: best setting for NYT; based on NPMI estimated with a 10-token sliding window over the reference corpus.

| | **WIKI** | **NYT** |
|---|---|---|
| G-LDA | $\sim 2$ minutes | $\sim 9$ minutes |
| D-VAE | $\sim 45$ minutes | $\sim 330$ minutes |
| ETM | $\sim 260$ minutes | $\sim 1300$ minutes |

Table 9: Runtimes for the three topic models on each of the two datasets. G-LDA requires CPUs only while the neural models use a single GPU. Compute resources detailed at the end of Section A.3.

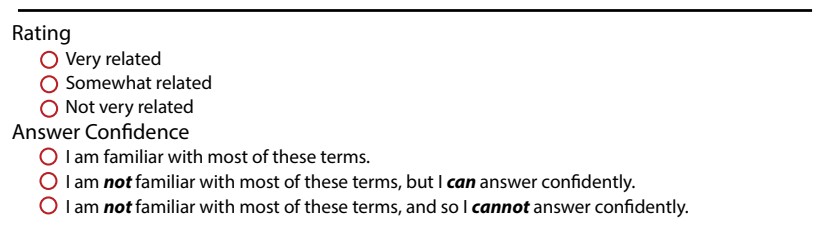

Figure 4: Ratings task presented to crowdworkers.

## A.5 Power Analysis for Human Evaluation Tasks

To select the number of crowdworkers, we conduct a power analysis with simulated data (Feiveson, 2002) by formulating a generative model of annotations (implementation included in released code). Card et al. (2020) find that many NLP experiments, including those relying on human evaluation, are insufficiently powered to detect model differences at reported levels.

**Word Intrusion.** Topic $k$ has a true latent binary label $z_k \sim \text{Bern}(0.5)$ ("coherent" or "incoherent") which indexes a parameter $p_{z_k} \in [0, 1]$. Annotator $i$ samples an answer to the intruder task $x_{ik} \sim \text{Bern}(p_{z_k})$. We therefore run a simulation of annotator data for two different models: MODEL A,

This survey asks you to look at lists of words produced by an automatic computer program. For each list, you'll be answering the question: "Which word doesn't belong?"

- You will be shown ten sets of six words.
- For each set, click the word whose meaning or usage is most unlike that of the other words.
- If you feel that multiple words do not belong, choose the one that you feel is most out of place.
- Do not base your decisions on how the word is pronounced or written or its grammatical function. For example, if you saw {apple, apricot, anvils, peach}, you would not choose "peach" because it doesn't start with "a", you would not choose "apricot" because it isn't five letters long, and you would not choose "apple" because it ends with a vowel. Ideally, you would choose "anvils" because it is not a fruit.

Here are some examples:

"baby", "crib", "diaper", "beer", "pacifier", "cry"

In this example the word 'beer' is the least related. All of the other words are closely related to each other, and related to infants.

Here is another, harder, example:

"Hard Drive", "motherboard", "video card", "processor", "RAM", "USB key"

While all of these terms are related to a computer, all but one of them are components inside of a computer. The best choice is therefore 'USB key'.

***You may not always know all the words and that's okay.***

This study should take approximately 10-15 minutes to complete. Your response will be completely anonymous.

This survey asks you to evaluate lists of words produced by an automatic method.

The computer model we are testing seeks to identify groups of words that are highly related to each other. You will be asked to select how related groups of words are on a 3-point scale.

The rating options are: Not Very Related, Somewhat Related, Very Related.

A helpful question to ask yourself is: "what is this group of words about?" If you can answer easily, then the words are probably related. Here is some guidance on how to apply these ratings and some examples.

***Very Related*** - Most of the words are clearly related to each other, and it would be easy to describe how they are related.

Example: "dog", "cat", "hamster", "rabbit", "snake" (An obvious way to describe the relationship here would be 'Pets')

Example: "brushwork", "canvases", "expressionism", "cubism", "modernism", "curators", "abstract_expressionism", "national_gallery_of_art", "museum", "fossils" (An obvious way to describe this would be "art", even though one or two of the words are not as clearly related to that.)

***Somewhat Related*** - The words are loosely related to each other, but there may be a few ambiguous, generic, or unrelated words

Example: "computer", "video", "new", "plug", "screen", "model" (In this example, some of the words are generic, and seem more closely related than others)

Example: "dog", "ball", "pet", "receipt", "pen" (In this example, some of the words seem closely related, but not all of them)

***Not Very Related*** - The words do not share any obvious relationship to each other. It would be difficult to describe how they are related to each other.

Example: "dog", "apple", "pencil", "earth", "computer"

This study should take approximately 10-15 minutes to complete. Your response will be completely anonymous.

(a)      (b)

Figure 5: Instructions for (a) word intrusion and (b) ratings

In this survey, the word lists are based on a computer analysis of The New York *Times*.

The New York *Times* is an American newspaper featuring articles from 1987 to 2007. Sections from a typical paper include International, National, New York Regional, Business, Technology, and Sports news; features on topics such as Dining, Movies, Travel, and Fashion; there are also obituaries and opinion pieces.

In this survey, the word lists are based on a computer analysis of Wikipedia.

Wikipedia is an online encyclopedia covering a huge range of topics. Articles can include biographies ("George Washington"), scientific phenomena ("Solar Eclipse"), art pieces ("La Danse"), music ("Amazing Grace"), transportation ("U.S. Route 131"), sports ("1952 winter olympics"), historical events or periods ("Tang Dynasty"), media and pop culture ("The Simpsons Movie"), places ("Yosemite National Park"), plants and animals ("koala"), and warfare ("USS Nevada (BB-36)"), among others.

(a)      (b)

Figure 6: Descriptions for (a) NYTimes and (b) Wikipedia.

which has a sample of $K = 50$ binary topic labels, $\boldsymbol{z}^{(A)}$; and MODEL B, with $r$ fewer "coherent" topics than A, $\sum_k z_k^{(B)} = \sum_k z_k^{(A)} - r$. After collecting pseudo-scores $\boldsymbol{x}^{(A)}$ and $\boldsymbol{x}^{(B)}$ for $M$ annotators, we run a one-tailed proportion test on the respective sums. The power is the proportion of significant tests over the total number of simulations $N$ (i.e., tests there where A is correctly determined to have higher scores than B). We set $p_0 = 1/6$ (chance of guessing), $p_1 = 0.85$ (roughly estimated with data from Chang et al., 2009).

**Ratings.** Rating scores on a 3-point scale are generated analogously, in a generalization of the above binary case. Assume that topics have true labels $\boldsymbol{z}_k \sim \text{Cat}(1/3, 1/3, 1/3)$. Annotator scores are noisy, so true labels are corrupted according to probabilities $p_{z_k} \in \Delta^2$. Here, MODEL A has a sample of $K = 50$ ratings on a 3-point scale. MODEL B has $r$ fewer 3-ratings ("very related") and $r$ greater 1-ratings ("not related") than A (the 2-ratings stay constant). After simulating scores for $M$ annotators for both "models," we run a one-tailed $U$-test (Mann and Whitney, 1947). Again, the power is the share of significant tests over all simulations $N$. Probabilities are $p_1 = [3/4, 1/4, 0]; p_2 = [1/4, 2/4, 1/4]; p_3 = [0, 1/4, 3/4]$, designed to roughly approximate empirical data—if we sample scores according to them and compute inter-"annotator" agreement, the one-versus-rest Spearman correlation is $\rho \approx 0.7$, or the same as the most-correlated dataset (NYT) in Aletras and Stevenson (2013) (our final data has $\rho = 0.75$).

For both settings, we set $r = 4$, the critical value $\alpha = 0.05$, and the desired power $1 - \beta = 0.9$. This analysis suggests fifteen annotators per topic for the ratings task and twenty-five for intrusion.

|  |  | NPMI (10-token window) | | | | $C_v$ (110-token window) | | | |
|---|---|---|---|---|---|---|---|---|---|
|  | Ref. Corpus → 
 Train Corpus ↓ | NYT | WIKI | Train | Val | NYT | WIKI | Train | Val |
| Intrusion | NYT | 2.42 | **4.16** | 2.11 | 1.97 | 2.50 | 3.27 | 2.55 | 2.40 |
|  | WIKI | 4.11 | 5.08 | **5.45** | 0.87 | 2.23 | 2.79 | 2.74 | 0.70 |
|  | Concatenated | 2.82 | **4.56** | 3.18 | 0.78 | 2.30 | 3.05 | 2.64 | 0.87 |
| Rating | NYT | 1.92 | 2.08 | 1.77 | 1.85 | 2.55 | 2.51 | **2.68** | 2.59 |
|  | WIKI | 2.97 | 4.10 | **4.29** | 1.45 | 2.01 | 2.82 | 2.86 | 0.80 |
|  | Concatenated | 2.20 | **2.75** | 2.52 | 1.17 | 2.27 | 2.60 | 2.74 | 1.07 |

Table 10: Logistic (intrusion) and ordinal probit (ratings) regression coefficients of automated metrics on human annotations. Underlined values have overlapping 95% confidence intervals with that of the **largest** value in each row.

### A.5.1 Power analysis for equivalence

To estimate the false discovery (omission) rates in Table 4, we need to determine when differences between human (automated) scores are not meaningful. Since human effects in the opposite direction of automated metrics also imply a false discovery, we conduct a test of non-inferiority; this is the same as using a large negative lower bound in the two-one-sided tests procedure for equivalence (Schuirmann, 1987; Wellek, 2010).

To determine the non-inferiority threshold—the bound $\epsilon$ below which we consider two sets of scores to be equivalent—we also conduct a power analysis, per the previous section. In this case, the simulation assumes *no* difference between the "true" labels of the model outputs, $z^{(A)} = z^{(B)}$. We estimate one-sided tests for each sample of human scores, with the null $H_0 : \mu_1^{(B)} - \mu^{(A)} > \epsilon$ for some bound $\epsilon$. We minimize $\epsilon$ while maintaining $\beta > 0.9$. This process produces $\epsilon = 0.05$ for the word intrusion task and $\epsilon = 0.11$ for the ratings task (roughly equivalent to a difference of 2.5 "incoherent" topics for both tasks, respectively).

For the automated scores, we generate two sets of scores $x_k \sim \mathcal{N}(0, \sigma^2)$; $\sigma^2 \sim \mathrm{Gamma}(\alpha, \beta)$ for $k = 1 \ldots K$ at each iteration, then conduct a t-test between each set. $\alpha$ and $\beta$ are selected such that the Gamma distribution approximately matches the empirical distribution of automated score variances. This leads to $\epsilon = 0.05$ for NPMI scores and $\epsilon = 0.06$ for the $C_v$ scores.

### A.6 Regression Results

Prior work (e.g., Röder et al., 2015) relates averaged human ratings to automated metrics using either Pearson or Spearman correlations. As an alternative that takes into account both the variation in human judgments as well as their numerical type, we estimate logistic and ordered probit regressions on the ratings and intrusion annotations, respectively. In Table 10, we report the estimated coefficients for each metric, finding that—on the whole—using the WIKI reference performs best, although the large estimated confidence intervals mitigate the strength of this conclusion.

### A.7 Filtering on Term Familiarity

Several topics, particularly those produced by D-VAE, contain terms that are not well-known to annotators (6.1). When a respondent is unfamiliar with a topic's words, their ratings for that topic may not accurately reflect its true coherence. For example, a mycologist may find the words in the fifth column of Table 1 highly related, whereas someone unfamiliar with fungi-related jargon may rate it poorly—indeed, the mean rating for this topic is 2.1 for those unfamiliar with terms and 2.6 for those who are familiar.

Since automated metrics do not take into account a term's familiarity to humans, we posit that automated metrics should be more predictive of human judgments among respondents who are familiar with topic terms. To test this hypothesis, we re-evaluate the relationships between automated metrics and human judgments *after removing* respondents who state they are not familiar with a topic's terms (Table 11). On the whole, results are much clearer than above; NPMI estimated using WIKI reference counts is strongly correlated across tasks and datasets. The false discovery rate is

| | Ref. Corpus → Train Corpus ↓ | NPMI (10-token window) | | | | $C_v$ (110-token window) | | | |
|---|---|---|---|---|---|---|---|---|---|
| | | NYT | WIKI | Train | Val | NYT | WIKI | Train | Val |
| Intrusion | NYT | 0.34 | 0.51 | 0.32 | 0.25 | 0.44 | **0.55** | 0.42 | 0.38 |
| | WIKI | 0.39 | 0.39 | 0.40 | 0.14 | 0.38 | **0.40** | 0.39 | 0.13 |
| | Concatenated | 0.36 | 0.45 | 0.36 | 0.18 | 0.41 | **0.48** | 0.41 | 0.26 |
| Rating | NYT | 0.45 | **0.59** | 0.44 | 0.43 | 0.51 | 0.58 | 0.53 | 0.52 |
| | WIKI | 0.45 | **0.51** | 0.51 | 0.21 | 0.44 | 0.51 | 0.51 | 0.23 |
| | Concatenated | 0.47 | **0.54** | 0.47 | 0.35 | 0.49 | 0.53 | 0.51 | 0.42 |

(a) Spearman correlation coefficients between mean human scores and automated metrics, compare to Table 3.

| | Ref. Corpus → Train Corpus ↓ | NPMI (10-token window) | | | $C_v$ (110-token window) | | |
|---|---|---|---|---|---|---|---|
| | | NYT | WIKI | Train | NYT | WIKI | Train |
| Intrusion | NYT | 53 / 55 | 46 / 50 | 56 / 52 | 41 / 34 | **28** / 27 | 42 / 33 |
| | WIKI | 38 / 76 | 36 / 77 | 32 / 76 | **29** / 37 | 31 / 43 | 33 / 39 |
| | Concatenated | 54 / 70 | 41 / 73 | 51 / 70 | 41 / 44 | **29** / 42 | 41 / 44 |
| Rating | NYT | 45 / 49 | 39 / 53 | 45 / 47 | 18 / 27 | **16** / 24 | 17 / 25 |
| | WIKI | 37 / 73 | 25 / 74 | 30 / 70 | 28 / 31 | 19 / 33 | **18** / 27 |
| | Concatenated | 45 / 64 | 38 / 68 | 42 / 64 | 26 / 36 | **21** / 36 | 27 / 33 |

(b) False discovery rate (1−precision, lower is better) and false omission rate of significant model differences when using automated metrics, compare to Table 4.

| | Ref. Corpus → Train Corpus ↓ | NPMI (10-token window) | | | | $C_v$ (110-token window) | | | |
|---|---|---|---|---|---|---|---|---|---|
| | | NYT | WIKI | Train | Val | NYT | WIKI | Train | Val |
| Intrusion | NYT | 3.71 | **7.14** | 3.04 | 2.54 | 3.34 | 4.54 | 3.23 | 2.94 |
| | WIKI | 5.87 | **6.46** | 6.19 | 0.85 | 3.23 | 3.59 | 3.39 | 0.42 |
| | Concatenated | 4.24 | **6.81** | 4.17 | 0.94 | 3.18 | 4.06 | 3.30 | 0.91 |
| Rating | NYT | 4.40 | **5.87** | 3.85 | 3.93 | 3.97 | 4.44 | 4.03 | 3.89 |
| | WIKI | 4.84 | **5.95** | 5.65 | 1.33 | 2.96 | 3.73 | 3.69 | 0.62 |
| | Concatenated | 4.49 | **5.80** | 4.56 | 1.78 | 3.45 | 3.91 | 3.81 | 1.32 |

(c) Logistic (intrusion) and ordinal probit (ratings) regression coefficients of automated metrics on human annotations, compare to Table 10.

Table 11: Tables 3, 4, and 10 after removing respondents who report a lack of familiarity with topic words.

lower overall, although automated metrics still misdiagnose significant results at a rate of one in six in even the best case.

These findings provide further evidence—per our discussion in Section 7—that future human evaluations of topic models ought to take into account domain expertise and information need.

## A.8 Five-point Ratings Scale

Although most prior work uses three-point scales for the ratingstask (Fig. 4), for comparison we also ask annotators to label the topic topic words with a five-point scale ranging from 1 ("not at all related") to 5 ("very related", no labels are given for points 2-4). Broadly, we find that values for correlations are reduced relative to the three-point scale (Table 12). We believe examining this discrepancy is an interesting direction for future work that re-visits human evaluation of topic models.

## A.9 Potential Negative Impact

Our work focuses its investigation on data from the English language alone. In this way, it further entrenches English-language primacy in NLP, and more crucially, findings may not translate directly to other languages. We caution the reader against applying claims made in this work to topic modeling

|  | Ref. Corpus → 
 Train Corpus ↓ | NPMI (10-token window) | | | | $C_v$ (110-token window) | | | |
|---|---|---|---|---|---|---|---|---|---|
|  |  | NYT | WIKI | Train | Val | NYT | WIKI | Train | Val |
| Rating (5-pt.) | NYT | 0.27 | **0.37** | 0.28 | 0.33 | 0.29 | 0.35 | 0.33 | 0.35 |
|  | WIKI | 0.15 | 0.21 | 0.29 | 0.43 | 0.10 | 0.16 | 0.17 | **0.50** |
|  | Concatenated | 0.21 | 0.30 | 0.28 | 0.32 | 0.20 | 0.26 | 0.26 | **0.39** |

Table 12: Spearman correlation coefficients between mean human scores for a **five-point** ratings scale (rather than three), compare to Table 3. Underlined values have overlapping 95% confidence intervals with that of the **largest** value in each row.

on corpora of other languages. It is even possible that one of the tasks designed to elicit human judgment (e.g., word intrusion) may not be amenable for use with other languages.

Concerning topic models more broadly, we note that others question the scholarly value of "distant reading" and the digital humanities in general (Marche, 2012; Allington et al., 2016). Do topic models encourage a passive, disengaged relationship to texts—fomenting conclusions about broad, generic trends rather than idiosyncratic specifics, leading us to miss the trees for the forest? As noted by Schmidt (2012), "topics neither can nor should be studied independently of a deep engagement in the actual word counts that build them." In this light, topic models can be viewed as an extension of the insidious neoliberal trend toward mass data harvesting that blurs differences between individuals and cultures. Researchers should take care to avoid such elisions when drawing conclusions from model outputs.

## A.10 NeurIPS Checklist