# OpenReview forum: "Is Automated Topic Model Evaluation Broken? The Incoherence of Coherence"
_NeurIPS.cc/2021/Conference — NeurIPS 2021 Spotlight_

### Official Review · Reviewer_KzLn · 2021-07-13

**Rating:** 7
**Confidence:** 4

**Summary:**

This paper questions the use of automatic metric, such as the ones based on NPMI, for evaluating the quality of topic models. It sets up a precise, reproducible framework (with code) that compare classic approaches (LDA) and recent ones based on neural networks (D-VAE). From my point of view, it is more a positioning paper that ca n help the community to change its habits than a novel solution to a given problem. However I think we need this type of papers which can lead to designing better models.

**Limitations And Societal Impact:**

I don't see any negative impact of this work.

**Main Review:**

Overall I found that the paper is well motivated and well written (despite a couple of typos, see below). I agree with the main assumptions: the current setting for evaluating the topic models is flawed and we need a stronger framework. It's not really surprising that recent neural models that leverage word embedding achieve better results when we look at metrics such as NPMI on an external corpus... what the authors point out when they mention the Goodharts’ law ("when a measure becomes a target, it ceases to be a good measure").

I'd add that judging the quality of topics should be based not only on the targeted task but also on the people who will use them. It seems clear to me that an expert will need more precise, focused topics than a non expert. We can find some discussion about this point in the paper (last paragraph of 6.1), but I think a novel evaluation setting may go beyond.

I will vote for accepting this paper because I think we need this kind of work in the community in order to face the so many topic models published every year. However I have a couple of small concerns I'd like to raise:

- I understand the choice to follow the original framework designed by the seminal papers (Chang et al., 2009; Lau et al., 2014). However, I think this work should have been the occasion to go a step beyond. For instance, the quality of topics may be studied at the light of the expertise / expectation of the user. Another possible improvement I saw in some previous work: it's not the same situation if the spurious word is taken from a close topic or a far topic.

- I think the final discussion should include the practical complexity of the different approaches. Table 11 of appendix is not included but it clearly shows that the "good old LDA" is considerably faster than modern NTM. If it's at the price of some points in accuracy, most of the users won't opt for NTM.

Some typos I found:
- a straightforward a generative model =>  a straightforward generative model
- uniquivocal => unequivocal
- into quesetion => into question

---

I've read authors' response and I'm still ok for accepting this paper.

**Time Spent Reviewing:**

2h

---

> ### Author Response · Authors · 2021-08-10
> **Response to Reviewer KzLn**
>
> Thanks for your review---we also hope this work will lead to a change within the community.
>
> Your point about computational cost of approaches is a very good one, especially since many (if not most) real-world users are running models locally and/or without access to a GPU. Space permitting, we will move the table with the run-time comparison from the appendix into the main paper (at the very least, we will add a reference). We will also include CPU wall-clock times for the NTM models for a fairer comparison.
>
> We discuss your idea about a novel evaluation setting in our general response.

---

### Official Review · Reviewer_Ti42 · 2021-07-14

**Rating:** 7
**Confidence:** 3

**Summary:**

The paper addresses standardization gap in automatic metrics used for evaluation of topic modeling as well as a validation gap for the coherence of those automatic metrics with the human judgement, both revisiting the models for which such validation had been done (LDA) as well as the recent Neural Top. Models which have been purely evaluated using the automatic metrics.

The authors provide a deep investigation on the automatic metrics currently in widespread use within Topic Modeling as well as a historic overview on the last studies of the coherence of the automatic metrics (NPMI) and human judgements.

One of the main contributions of the paper is the rigorous study and comparison of the human judgements and the automatic metrics NPMI and C_v ensuring the statistical power of the test (involving large-scale evaluation by crowdsourced judges) making this work a steady reference point for future works.

**Limitations And Societal Impact:**

limitations addressed ; societal impact doesn't apply

**Main Review:**

### significance

The work belongs to the rare kind of "sanity check" works around the evaluation of the modern automated methods, in this, case, topic modeling. I believe it has a potential not only affect future work on topic modeling and how it's evaluated but also will inspire thoughts on careful analysis of the adequacy of automated metric used in other fields.

The work is valuable for the community. It also sets quite a high standard on the statistical rigor of experimentation.

### originality

It is not (sufficiently) often that the research community undertakes an effort of verifying fundamentals, in this case of an automated metric used in topic modeling. I consider the paper sufficiently original.

### clarity

Given the complex nature of experiments, the paper successfully explains the nature of the problem, the proposed approach, and the experimentation procedure. Additionally, a discussion on the consequences of the paper is included in Section 7.

Minor observations:

In **4.2 Models**:

"estimated using the reference corpus (Table 6) " - Table 6 is in Appendix and lists the referenced work, so not quite related to the reference corpus. Might be some confusion.

"...this study is not focused on efficiency of the implementation Ethayarajh and Jurafsky (though cf. 2020)." -> reference not quite clear

Footnote 9: should it also provide a GitHub link?


In **5 Human Evaluations of Topic Quality**:

Within main text body: "we include an additional sixteen synthetic poor-quality topics"
The footnote: "Eight topics each are generated from the vocabularies of NYT and WIKI." - so, why only 8 topics out of 16 are explained?


### quality

A high quality writing, very few typos.

LANGUAGE

A few typos:

In **5 Human Evaluations of Topic Quality**:

"Adopting a straightforward a generative model " -> extra "a"

"contents can be esoteric (e.g.,last columns of Table 1)," -> I'd suggest replacing the word "esoteric" in general (more so when the indicated example is about nuclear physics research project).

Figure 1 caption: "orange circle" -  colors are not transmitted at black-and-white printing. It's always a great courtesy to the reader to use grayscale in graphics.

Figure 1 caption: "topics derived from on Wikipedia." -> too many prepositions

In **6 Human Judgment Differs From Automated Metrics** :

"we compare and contrast model comparisons" - sounds like tautology

In **6.1 Human Assessment**:

"However,D-VAE" -> space

In **6.2 Automated Metrics**:

"per task (table 4)." -> Table (and a few other references to tables, sections, and figures on the same page)

"in the the NTM literature" -> extra "the"


REFERENCES


1. Aren't these 2 references the same paper?:

* Jesse Dodge, Suchin Gururangan, D. Card, Roy Schwartz, and Noah A. Smith. 2019a. Show your work: Improved reporting of experimental results. ArXiv, abs/1909.03004.

* Jesse Dodge, Suchin Gururangan, Dallas Card, Roy Schwartz, and Noah A. Smith. 2019b. Show your work: Improved reporting of experimental results. In Proceedings of Empirical Methods in Natural Language Processing. Association for Computational Linguistics.

2. It's advisable to brush up the reference formatting: some first names are abbreviated, most are not; sometimes "." is omitted (e.g., "David M Blei." -- after "M")

**Time Spent Reviewing:**

4

---

> ### Author Response · Authors · 2021-08-10
> **Response to Reviewer Ti42**
>
> Thank you for your thorough and positive comments, especially that you feel our work can serve as a reference point going forward---and that you appreciated the statistical rigor of our work.
>
> We will fix all noted typos and references in the final version. Thank you very much for taking the time to point these out.
>
> Regarding synthetic low-quality topics: if we understood your question correctly, we have 8 topics for each of the two datasets, totalling 16.

---

> > ### Comment · Reviewer_Ti42 · 2021-08-25
> > **Best of luck!**
> >
> > Dear Authors,
> > Thank you for your response.
> >
> > I believe my initial score is well-deserved and adequate , so I'll keep it.

---

### Official Review · Reviewer_8PGF · 2021-07-16

**Rating:** 7
**Confidence:** 4

**Summary:**

Topic model evaluation in terms of the quality of learned topics is an important problem in topic modelling, the validation of which has a profound impact on the model development and its application in cross disciplines. This paper reassesses the automatic topic model evaluation metrics, e.g., topic coherence scores like NPMI, that are commonly adopted in the topic modelling literature. It points out the potential flaws of applying those measures to neural topic models developed recently, via conducting a set of human evaluations. The paper identifies two gaps existing in the current evaluation paradigm used in almost all NMTs, the validation gap and the standardisation gap through analysing a large number of NTMs. Its findings could have wide implications in the practice.

**Limitations And Societal Impact:**

The findings of this paper suggest that the automatic topic evaluation should be carefully reconsidered and reassessed for newer models, like those recently developed neural topic models. They are going to have an impact on the adaptation of topic models in analyzing corpora in different domains.

**Main Review:**

* The paper is well motivated. The meta-analysis of the forty neural topic models is a substantial amount of work. The authors find, for example, inconsistency in document pre-processing, a lack of clarity in the calculation of topic coherence scores, and the ambiguity of comparison settings, which highlights potentially the problems for topic modelling practitioners. I like the argument on their skepticisim on the correlation found on classical models may not be applied for neural topic models, which seems to be supported by their human evaluation.
* In regard to the human evaluation paradigm, there is nothing new but adopting similar evaluation approaches used by previous work on human evaluation of topic models, i.e., topic quality rating and word intrusion. The difference is that the authors applied human evaluation to neural topic models.
* The authors carry out both human and automatic evaluation on topics generated by one classical model, Gibbs-LDA and two neural topic models, D-VAE and ETM, and find that “human judgment differs from automatic metrics”, which seems to support their skepticism on applying existing topic coherence measures to neural topic models.  However, to make the finding more convincing, should one consider more topic models in the experiment in order to draw such a conclusion? It would be good to see this finding is widely applicable to NTMs.
* The paper is written pretty well, and the necessary details of the experiments are provided, including the screenshots of instructions, the hourly wage for the participants in either the main paper or the supplementary materials. Given that there are human subjects involved, one might wonder if the ethics are sought and approved.

Some other minor comments
* In section 6.1 human assessment, what is the definition of “sub-optimal topics”?
* The discussion of Table 3, it seems that the words for “good topics” and “bad topics” are not reflected in the table.

----After the author's response----
I acknowledge that the authors' response to the review comments has been read and considered in the rating. I would like to keep my current rating.
---

**Time Spent Reviewing:**

>5

---

> ### Author Response · Authors · 2021-08-10
> **Response to Reviewer 8PGF**
>
> Your review is much appreciated. We are particularly pleased that you feel our findings could have wide implications for topic modeling (and we certainly hope so too)!
>
> We agree that additional models could be useful, but we decided to allocate our budget toward collecting more human evaluations on fewer models to ensure reasonable statistical power. As a result, our results clearly support the primary conclusion: automated coherence metrics are inadequate for model comparison. Additional models wouldn’t invalidate this conclusion. Also note that this is the same number of models as Chang et al. (2009) (and hence Lau et al. 2014).
>
> Also note that we selected NTMs that are representative of the field: ETM uses word embeddings, and our D-VAE implementation is structurally very similar to the most popular NTMs (ProdLDA, Scholar, etc.), and is nominally “state-of-the-art” for NPMI. Anecdotally, these other NTMs appear to suffer from the same problems, producing similar topics to D-VAE. That said, we agree that evaluating other models should be included in future work, e.g., the development of new metrics.
>
> Regarding ethical and IRB-related questions associated with crowdsourcing---which we very much agree should be asked!---the crowdworker activities in our study do not meet the definition of human subjects research, so we did not require approval from an IRB (or equivalent body). The crowdworkers themselves are not the object of study; rather, they were hired to provide information (annotations) as a paid service for purposes of developing or improving a technology. As a final note, we strongly believe that ethical considerations are important whether or not IRB approval is involved. For crowdsourcing a key question is fairness of compensation; we paid workers a base rate of $15 per hour (in practice it was typically higher; see appendix for details). We are happy to discuss any of this reasoning further if that is of interest.
>
> To answer your minor comments:
> * Sub-optimal topics are defined in footnote 11, but we refer to them with a different name (“poor-quality topics”). We will make the language consistent in the final version.
> * Thanks for pointing out the issue with our example topics. We likely updated the table and neglected to change the corresponding text, which we will correct.

---

### Official Review · Reviewer_KzUd · 2021-07-18

**Rating:** 7
**Confidence:** 3

**Summary:**

This paper investigates automated evaluation metrics of topic models in terms of coherence such as NPMI. NPMI is widely used to evaluate the performance of topic models, but the authors tackle that it is less related to human evaluation metrics such as rating and intrusion. For experiments, the authors train three topic models, LDA, D-VAE, and ETM with two different corpora, Wikipedia and new york times. To evaluate the topics by humans, the authors ask at least fifteen crowdworkers per topic to avoid insufficient power of human evaluation problem. Experiments show that D-VAE outperforms other models in terms of NPMI, but there are no significant differences among models by human evaluations.

The main strength of this paper is claiming the usefulness of automated evaluations of topic models with deep considerations that follows recent concerns of NLP research evaluations. I really enjoy reading this paper since it can guide the way to compare the two or more different evaluation metrics in various domains including topic modeling.


**Limitations And Societal Impact:**


I have some questions and comments that I would like to hear from the authors.

- How about rethinking about the human evaluation methods [1]? The authors claim the automated evaluation methods by comparing with the human judgments. What if the human evaluation methods are not good? I am curious and doubtful about the following: Can we capture the difference of relatedness of words by only three scales? What if we increase the numbers to five or ten (i.e. five Likert scales)? Can we possibly change the difficulty of the intrusion task? Of course, this is out of the scope of this paper, but I am curious about the author's opinions for better automated topic model evaluations.
- Recent papers suggest various automated topic model evaluation methods that are better than NPMI [2,3]. It would be better to discuss these.
- Why is the Wikipedia corpus best correlated with human scores even though the training data is not the same? It would be better to analyze the reasons such as showing the examples.

[1] Clark E, August T, Serrano S, Haduong N, Gururangan S, Smith NA. All That's' Human'Is Not Gold: Evaluating Human Evaluation of Generated Text. In Proceedings of the ACL 2021

[2] Lund J, Armstrong P, Fearn W, Cowley S, Byun C, Boyd-Graber J, Seppi K. Automatic Evaluation of Local Topic Quality. In Proceedings of the ACL 2019

[3] Xing L, Paul M, Carenini G. Evaluating Topic Quality with Posterior Variability. In Proceedings of the EMNLP 2019


**Main Review:**

Originality: Tasks, models, and metrics are not new, but the analysis methods are somewhat new.

Quality: Experiments and methods are sound. Results are quite interesting but it would be better to show the reason for results that are not the same as previous work (i.e. why the Wikipedia corpus is best correlated with human scores even the training data is not the same)

Clarity: This paper is well-written, just some typos and hyperlinks need to be fixed. For example, ‘(Table 6)’ in page 6 will be ‘(Table 3)’.

Significance: The claim is important because we need to create better automated evaluation methods for topic models. And the final section is also good for discussion about how to create a better automated topic modeling evaluation metric.


**Time Spent Reviewing:**

10

---

> ### Author Response · Authors · 2021-08-10
> **Response to Reviewer KzUd**
>
> Thank you for your thoughtful and positive feedback! We are especially glad you feel that our statistical framework for comparing human and automated evaluations is general, and indeed we hope that others adapt it for use in other areas (e.g., machine translation).
>
> To respond to your questions: first, we agree that using other human evaluations, like a five-point Likert scale, would be useful. Indeed, we debated internally whether a five-point scale would be more appropriate. However, a three-point scale is used in all prior work. Ultimately, we felt that remaining consistent with previous efforts was most important, since we wanted to replicate these experimental conditions under the neural topic model setting (and with well-powered tests). If accepted, we will run a small pilot study with a five-point scale and include a discussion of the results in the appendix, since this is of interest to us as well. We comment on your note about introducing alternative types of human evaluations in our general response.
>
> We are aware of the other automated evaluations in the papers you referenced, and entirely agree that the field should be expanding its use of alternative metrics. Doogan and Buntine (2020), as we mention in our paper, have pointed out the need for metrics better suited to real-world applications. We will expand on the need for other topic metrics further in our discussion in the final version.
>
> That said, a primary focus of our work is topic coherence for system evaluation (e.g., NPMI), because it remains the _de facto_ standard in topic modeling development papers (as shown by our meta analysis). We are very familiar with Lund et al 2019, but this is applicable for situations when token-level assignments are of interest and is therefore orthogonal to topic-level evaluations like NPMI. Likewise, Xing et al. 2019 narrowly rely on the dynamics of Gibbs sampling. This makes their approach an excellent diagnostic for models with those inference strategies but it does not generalize to other model types. Furthermore, neither paper addresses the key question of system evaluation: they each appear to use a single model. Any claims they make about better measurements may not apply to neural models. While conducting these evaluations is valuable, it is not the goal of our work.
>
> Finally, there are several possible explanations for why Wikipedia shows somewhat better correlations with human judgments than Lau et al. (2014)---differences in underlying reference corpora, sets of models, and numbers of participants. One reason may be that absolute differences between the Wikipedia and NYT correlations in that study aren’t terribly consistent to begin with: at the topic level, they are 0.741/0.722 for the ratings task; 0.612/0.648 for word intrusion. That said, if we find any particularly illustrative examples that help explain the correlations in our data, we will include them.

---

> > ### Comment · Reviewer_KzUd · 2021-08-22
> > **Thank you for the responses**
> >
> > Thank you for your responses to my questions.
> >
> > I agree that this paper focuses on the limitations of the existing and popular automatic evaluation metrics - NPMI. And the authors promise that they will include the discussion and more results in the paper. I've also read other comments and responses too.
> >
> > Finally, I've changed the total score.

---

### Author Response · Authors · 2021-08-10
**General Response to Reviewers**

Thanks to all reviewers for their constructive and positive feedback. We are very pleased that they believe this work can serve as an important touchstone for the field moving forward. In addition, we are glad they feel our methods are sufficiently general to apply to the study of automated metrics in other fields. We hope this turns out to be the case! Reviewers also remarked on the clarity of presentation and the rigor of the experiments, as well as the value of our meta-analysis.

Two reviewers expressed an interest in seeing novel human evaluations, other than those presented here (although acknowledged that using the existing framework was understandable). We entirely agree that newer human evaluations are necessary to move the field forward and inspire better automated metrics---we especially feel that evaluations should be tailored to likely use-cases for topic models, which usually involves domain-experts.

Despite our commitment to this idea, we did not include them here because we felt they were out of scope for the focused discussion in this paper. Showing systemic flaws in the existing paradigm creates the groundwork for new evaluation strategies, namely, ones that are sensitive to intended use cases. We thought the paper would have greater impact and relevance if we focused on making the strongest and best supported argument using the framework currently in use by the majority of people.

In addition, more comprehensive human evaluations require the assessment of more than just the top words in each topic (e.g., the works cited by [reviewer 1](https://openreview.net/forum?id=tjdHCnPqoo&noteId=i9sXur4XUx), as well as Doogan and Buntine, 2020). However, this shifts the attention away from topic coherence, the primary subject of our work, and the quantity receiving the most attention in the literature.

In the final version, we will expand on this argument, as well as propose its implications for future work. We will further discuss the results from our “approximation” of domain-expertise in this context (using the word-knowledge question). In some ongoing work, we have already begun to study differences between experts and laypeople.

---

> ### Comment · Program_Chairs · 2021-08-27
> **Your comment is not visible to the Area Chair**
>
> Dear Authors,
>
> We noticed you did not include the Area Chair handling your paper in the Readers field for this comment.  Please edit the Readers field to include the Area Chair.  Otherwise, OpenReview will not allow them to see the comment.
>
> Thanks,
> Program Chairs

---

### Decision · Program_Chairs · 2021-09-27

**Decision:**

Accept (Spotlight)

**Comment:**

All reviews recommended acceptance. The work was seen as somewhat novel and well motivated for an important task of topic model evaluation, and experiments were considered sound. Thus the work seems clearly valuable to the community. Some questions were raised about the experiments (analysis why the Wikipedia corpus is best correlated, alternatives to NPMI, use of only three-point scales in human evaluations, considering expertise of the user), but overall the work seems clearly acceptable.